# Clinical pneumonia in the hospitalised child in Malawi in the post-pneumococcal conjugate vaccine era: a prospective hospital-based observational study

Pui-Ying Iroh Tam [1,2,3] James Chirombo,[4] Marc Henrion,[2,4] Laura Newberry,[1] Ivan Mambule,[3,5] Dean Everett,[3,6] Charles Mwansambo,[7] Nigel Cunliffe,[5] Neil French,[8] Robert S Heyderman,[9] Naor Bar-Zeev,[1,3,9,10] For the VacSurv Consortium

PI and JC are joint first authors.

For numbered affiliations see end of article.

**Correspondence to**
Dr Pui-Ying Iroh Tam;
irohtam@mlw.mw

## ABSTRACT

**Objective** Assess characteristics of clinical pneumonia after introduction of pneumococcal conjugate vaccine (PCV), by HIV exposure status, in children hospitalised in a governmental hospital in Malawi.

**Methods and findings** We evaluated 1139 children ≤5 years old hospitalised with clinical pneumonia: 101 HIV-exposed, uninfected (HEU) and 1038 HIV-unexposed, uninfected (HUU). Median age was 11 months (IQR 6–20), 59% were male, median mid-upper arm circumference (MUAC) was 14 cm (IQR 13–15) and mean weight-for-height z score was −0.7 (±2.5). The highest Respiratory Index of Severity in Children (RISC) scores were allocated to 10.4% of the overall cohort. Only 45.7% had fever, and 37.2% had at least one danger sign at presentation. The most common clinical features were crackles (54.7%), nasal flaring (53.5%) and lower chest wall indrawing (53.2%). Compared with HUU, HEU children were significantly younger (9 months vs 11 months), with lower mean birth weight (2.8 kg vs 3.0 kg) and MUAC (13.6 cm vs 14.0 cm), had higher prevalence of vomiting (32.7% vs 22.0%), tachypnoea (68.4% vs 49.8%) and highest RISC scores (20.0% vs 9.4%). Five children died (0.4%). However, clinical outcomes were similar for both groups.

**Conclusions** In this post-PCV setting where prevalence of HIV and malnutrition is high, children hospitalised fulfilling the WHO Integrated Management of Childhood Illness criteria for clinical pneumonia present with heterogeneous features. These vary by HIV exposure status but this does not influence either the frequency of danger signs or mortality. The poor performance of available severity scores in this population and the absence of more specific diagnostics hinder appropriate antimicrobial stewardship and the rational application of other interventions.

## Strengths and limitations of this study

► We evaluated over 1100 children hospitalised with pneumonia in a low-income country setting after introduction of pneumococcal conjugate vaccine (PCV).

► This observational cohort was nested within a prospective hospital-based study of PCV13 effectiveness.

► We assessed the demographic and clinical characteristics of clinical pneumonia patients and compared HIV-exposed, uninfected versus HIV-unexposed, uninfected children and computed Respiratory Index of Severity in Children scores for severe pneumonia.

► This study was limited by the low mortality rate, small proportion of HIV-exposed infants with HIV status tested by PCR, the single-centre study, a substantial proportion of incomplete files and lack of recruitment of moribund children who died very soon after arrival into the study. However, this was a detailed prospective study conducted in a low-resource setting in the post-PCV era where robust diagnostic testing was available.

HIV prevalence among pregnant women is 10.8%,[1] and where a high burden of malaria and malnutrition (stunting prevalence of 39%[2]) contributes to an under-5 mortality rate of 55 per 1000 live births.[3]

WHO Integrated Management of Childhood Illness (IMCI) guidelines for community and referral level management of children with pneumonia rely on clinical recognition of features of severe disease. WHO lists general danger signs, such as inability to drink, persistent vomiting, convulsions, lethargy or unconsciousness, or stridor in a calm child, which are clinical features used to

## INTRODUCTION

Pneumonia is the leading cause of morbidity and mortality in children globally. This disease is of particular importance in low-income countries such as Malawi, where

identify children who should be treated in an inpatient setting and to receive injectable antibiotics.[4] However, clinical features are non-specific and can reflect a variety of other infectious (eg, bronchiolitis, malaria, sepsis, tuberculosis, pneumocystis) and non-infectious (eg, asthma, lymphocytic interstitial pneumonitis, pulmonary Kaposi sarcoma) conditions. Inappropriate and overuse of antimicrobials in low resource settings where antimicrobial prescribing is based more on clinical features than objective diagnostic data has implications for antimicrobial stewardship in the post-pneumococcal conjugate vaccine (PCV) era, particularly in settings where antimicrobial choice and supply is limited and antimicrobial resistance rates are rising.[5]

Few risk scoring systems to predict severe pneumonia exist that are adaptable to low-income countries with high HIV prevalence—Respiratory Index of Severity in Children-Malawi (RISC-Malawi) is the only score from children from a low-income country, and did not consider HIV status. Furthermore, although the alternative, RISC score does include a scoring system based on HIV infection, no studies have specifically evaluated whether HIV exposure status affects the presence and predictive value of clinical features for severe disease and case-fatality. As HIV exposed, uninfected (HEU) children constitute a growing population and, compared with HIV-unexposed, uninfected (HUU) remain vulnerable to more severe pneumonia, more often fail empiric pneumonia treatment, and have higher rates of hospitalisation and death than HUU children,[6–9] this population is of growing clinical relevance. Our objectives were to describe the clinical features of children hospitalised with pneumonia, calculate risk scores for our population, and to determine whether differences in clinical characteristics and calculated risk scores exist by HIV exposure status.

## METHODS

This observational cohort was nested within a prospective hospital-based study of 13-valent PCV effectiveness conducted at Queen Elizabeth Central Hospital between 2 April 2013 and 3 August 2016. Children born on or after 1 October 2012 (based on verbal and written documentation from parents on time since birth) and therefore eligible for PCV and admitted with a diagnosis of clinical pneumonia were enrolled (online supplemental table 1). Children were excluded if they had known oncological or congenital heart disease, were admitted ≥48 hours before recruitment, were re-admitted to hospital within 14 days of a previous hospitalisation, or presented moribund with impending death. Study staff performed active case finding Monday–Friday during daytime in the emergency department and inpatient wards, and followed up study participants 6 weeks following hospital discharge with a home visit or phone call to confirm vital status.

With informed written parental consent, all enrolled children had baseline characteristics measured which included clinical, laboratory and sociodemographic

variables. Oxygen saturation was measured with an appropriately sized sensor and a Nellcor pulse oximeter (Welch Allyn Spot Vital Signs Devices, Skaneateles Falls, New York, USA) in room air. Children with oxygen saturations <90% in room air were given supplemental oxygen via nasal cannula using an oxygen concentrator. At clinician decision and if a free circuit was available, children with severe respiratory distress, signs of exhaustion or worsening hypoxaemia received bubble continuous positive airway pressure (Pumani, third Stone Design, San Rafael, California, USA). Blood was routinely collected for packed cell volume and malaria parasite thick film. Mothers and infants were tested for HIV according to the Malawi National HIV Guidelines.[10] A blood culture was obtained only when the admitting clinician suspected sepsis. Children with suspected meningitis, malaria or anaemia, were managed according to respective WHO and hospital treatment guidelines.[11]

### Patient and public involvement

Patients or the public were not involved in the design, or conduct, or reporting, or dissemination plans of our research.

### Case definition

Pneumonia was diagnosed clinically per WHO IMCI criteria.

### Statistical analysis

We assessed the demographic and clinical characteristics of clinical pneumonia patients and compared HEU versus HUU children using Fisher's exact test, two-sample t-test or Wilcoxon rank-sum tests, respectively, for categorical, continuous normal and continuous skewed variables.

We computed RISC scores for severe pneumonia in our cohort,[12–14] using the variables listed in table 1. We had insufficient variables to compute modified RISC (mRISC) and RISC-Malawi scores.[13]

### RESULTS

A total of 1660 children were recruited into the study, among whose files were incomplete for 521. We included 1139 children in the analysis, of whom 101 were HEU and 1,038 HUU. The median age was 11 months (IQR 6.0–20.0), 59% were male, and the median mid-upper

**Table 1** List of variables used to compute RISC scores

| RISC | Score |
|---|---|
| Oxygen saturation ≤90% | 3 |
| Chest indrawing | 2 |
| Refusal to feed | 1 |
| Wheezing | -2 |
| WHO weight for age z-score ≤ −3 | 2 |
| WHO weight for age z-score −3 ≤_z < −2 | 1 |

RISC, Respiratory Index of Severity in Children.

arm circumference (MUAC) was 14 cm (IQR 13–15) and mean weight-for-height z score −0.7 (±2.5; table 2 and online supplemental table 2).

In our cohort, no single clinical feature was present at enrolment in more than 55% of the population: the most common clinical features were crackles (54.7%), nasal flaring (53.5%) and lower chest wall indrawing (53.2%). Less than half had fever (45.7%), and at least one IMCI danger sign (37.2%). The median oxygen saturation at presentation was 95% (IQR 89–98), with 25.4% having oxygen saturations below 90%. Among this cohort, 33.9% received oxygen and 2.1% were placed on continuous positive airway pressure (CPAP). The highest RISC scores (3+) were allocated to 10.4% of the overall cohort, respectively. Five children died (0.4%).

Ten per cent of children in our cohort were HEU, and notable differences between HEU and HUU were observed in several areas. In the HEU group, children were younger (median age 9 months (IQR 5.0–15.5) vs 11 months (IQR 6.0–20.0), p=0.02), with lower birth weight (median 2.8 kg (IQR 1.8–3.2) vs 3.0 kg (IQR 2.4–3.3), p=0.03), lower weight-for-height z score (mean −1.2±2.5 vs −0.6±2.5, p<0.01) and lower MUAC (median 13.6 cm (IQR 12.6–14.6) vs 14 cm (IQR 13–15), p<0.01). Compared with the HUU group, HEU children had significantly higher prevalence of vomiting (32.7% vs 22.0%, p=0.02) and age-defined tachypnoea in infancy (68.4% vs 49.8%, p=0.02). More HEU children presented with at least one IMCI danger sign (46.5% vs 36.3%, p=0.05). We did not observe significant differences in respiratory symptoms and in proportions of children who died between groups, although HEU children received oxygen at a significantly higher rate than HUU children (44.0% vs 32.9%, p=0.04). Clinical outcomes between the HEU and HUU groups were also similar (table 2). However, RISC scores were significantly different between HIV exposure status, with 20.0% of HEU with the highest scores, compared with 9.4% of HUU (figure 1). The risk score had a non-parametric distribution and many subjects scored 0.

## DISCUSSION

In our prospective cohort of children presenting with WHO IMCI criteria for clinical pneumonia in the post-PCV era, the most common clinical features were crackles, nasal flaring and lower chest wall indrawing. Despite the clinical diagnosis of pneumonia, features were heterogeneous. Thirty-four per cent had an oxygen requirement and 2.1% needed CPAP support, though with best available treatment, overall mortality was low—and possibly a reflection of improvements in child health overall as a result of antiretroviral and antimalarial therapies, vaccination and community management of nutrition. Ten per cent of children hospitalised with pneumonia were HEU and this group was significantly younger, with lower birth weight and anthropometric parameters. HEU children had a significantly higher prevalence of vomiting,

age-defined tachypnoea in infancy, and higher RISC scores, compared with HUU children.

Though HEU children were significantly younger and had significantly lower MUAC than HUU children, and therefore, age and anthropometry are potential triage indicators for use in the clinical management of pneumonia, these differences were subtle and may not be useful for a clinician, who in low-resource settings make decisions primarily on clinical suspicion. WHO IMCI clinical criteria for pneumonia are non-specific, and based on presence of cough, fast and difficult breathing. The non-specific nature of the diagnosis, which covers both infectious and non-infectious causes, is deliberately intended to increase sensitivity; however, as reflected in this study, this may have resulted in the term being used with a variety of features. Much research has been expended on developing pneumonia risk scores to identify children at risk of severe disease. However, the limitations of existing pneumonia risk scores for children in low-income and middle-income country settings,[15] including Pneumonia Etiology Research in Child Health (PERCH),[16] Respiratory Syncytial Virus Network,[17] RISC,[14] mRISC[12] and RISC-Malawi,[13] is that the predictive capacity is generally modest and, in the case of PERCH, was found to be not as discriminatory as the WHO danger signs. The risk scores we calculated in our cohort were not normally distributed and yielded surprisingly divergent results, indicating that they are limited to their study populations and are not generalisable.

In our cohort, the lack of predominating features of clinical pneumonia points to another possibility that, in post-PCV settings where HIV and chronic malnutrition rates are high (10% and 39%, respectively), children do not present with typical signs and symptoms. The difference in distribution of risk scores within our cohort highlights the fact that risk scores are population-specific and cohort-specific. In our cohort a substantial proportion of children met criteria for severe acute malnutrition (9.7% had weight-for-height z scores ≤3; and 4.1% had MUAC <11.5 cm), indicating how widespread chronic malnutrition is in Malawi. This pervasive underlying comorbidity leads to subsequent issues with identification and management of disease; and may in part explain overuse of antimicrobials in this setting, one factor contributing to an increase in antimicrobial resistance.

Antimicrobial stewardship and infection control issues in low-resource settings have been particularly problematic among young children, with significant increases in antimicrobial resistance noted among young infants with bloodstream infections over a 20-year period,[5] and concerns over resistance rates in childhood pneumonia in sub-Saharan Africa.[18] Challenges with diagnostic capacity and the limitations of clinical diagnosis hinder appropriate antimicrobial stewardship.[19] Risk scores that are adequately discriminatory, therefore, present an opportunity for targeted interventions, including early aggressive support, as well as judicious allocation of resources.

**Table 2** Characteristics of children hospitalised with clinical pneumonia, stratified by HIV exposure status

| Characteristics | All (N=1139) | Group | | P value |
| --- | --- | --- | --- | --- |
| | | HEU (N=101) | HUU (N=1038) | |
| Demographic and physiological characteristics | | | | |
| Age, months (m) | | | | 0.02* |
| N | 1118 | 99 | 1019 | |
| Median (IQR) | 11.0 (6.0–20.0) | 9.0 (5.0–15.5) | 11.0 (6.0–20.0) | |
| Sex | | | | 0.92† |
| N | 1139 | 101 | 1038 | |
| Male (%) | 677 (59.4) | 61 (60.4) | 616 (59.3) | |
| Birth weight, kg | | | | 0.03* |
| N | 1117 | 97 | 1020 | |
| Median (IQR) | 3.0 (2.3–3.3) | 2.8 (1.8–3.2) | 3.0 (2.4–3.3) | |
| Weight for height | | | | <0.01‡ |
| N | 1098 | 100 | 998 | |
| Mean (SD) | −0.7 (2.5) | −1.2 (2.5) | −0.6 (2.5) | |
| MUAC, cm | | | | <0.01* |
| N | 1129 | 101 | 1028 | |
| Median (IQR) | 14.0 (13.0–15. 0) | 13.6 (12.6–14.6) | 14.0 (13.0–15.0) | |
| Temperature | | | | 0.02* |
| N | 1133 | 100 | 1033 | |
| Median (IQR) | 37.8 (36.9–38.6) | 38.0 (37.2–39. 0) | 37.8 (36.8–38.5) | |
| Fever | | | | 0.06 |
| N | 1133 | 100 | 1033 | |
| Yes (%) | 518 (45.7) | 55 (55.0) | 463 (44.8) | |
| Antibiotic use | | | | 0.34† |
| N | 1131 | 101 | 1030 | |
| Yes (%) | 451 (39.9) | 45 (44.6) | 406 (39.4) | |
| Inability to drink | | | | 0.08† |
| N | 1134 | 101 | 1033 | |
| Yes (%) | 91 (8.0) | 13 (12.9) | 78 (7.6) | |
| Vomiting | | | | 0.02† |
| N | 1137 | 101 | 1036 | |
| Yes (%) | 261 (23.0) | 33 (32.7) | 228 (22.0) | |
| At least one IMCI danger sign | | | | 0.05† |
| N | 1139 | 101 | 1038 | |
| Yes (%) | 424 (37.2) | 47 (46.5) | 377 (36.3) | |
| Clinical features | | | | |
| Nasal flaring | | | | 0.92† |
| N | 1137 | 101 | 1036 | |
| Yes (%) | 608 (53.5) | 55 (54.5) | 553 (53.4) | |
| Lower chest wall indrawing | | | | 1.00† |
| N | 1137 | 101 | 1036 | |
| Yes (%) | 605 (53.2) | 54 (53.5) | 551 (53.2) | |
| Stridor | | | | 0.44† |
| N | 1137 | 101 | 1036 | |
| Yes (%) | 22 (1.9) | 3 (3.0) | 19 (1.8) | |
| Age-defined tachypnoea | | | | 0.02† |
| N | 666 | 57 | 609 | |

Continued

**Table 2** Continued

| Characteristics | All (N=1139) | Group | | P value |
| --- | --- | --- | --- | --- |
| | | HEU (N=101) | HUU (N=1038) | |
| Age 0–12 months and >50 breaths/min | 342 (51.4) | 39 (68.4) | 303 (49.8) | |
| Age 12–60 months and >40 breaths/min | 324 (48.7) | 18 (31.6) | 306 (50.3) | |
| Oral thrush/sores | | | | 0.11† |
| N | 1138 | 101 | 1037 | |
| Yes (%) | 21 (1.9) | 4 (4.0) | 17 (1.6) | |
| Hypoxaemia <90% | | | | 0.40† |
| N | 1128 | 100 | 1028 | |
| Yes (%) | 286 (25.4) | 29 (29.0) | 257 (25.0) | |
| Risk scores | | | | |
| RISC | | | | <0.01† |
| N | 1120 | 100 | 1020 | |
| –2 to 2 | 1004 (89.6) | 80 (80.0) | 924 (90.6) | |
| 3–6 | 116 (10.4) | 20 (20.0) | 96 (9.4) | |
| Clinical management | | | | |
| Received oxygen | | | | 0.04† |
| N | 1129 | 100 | 1029 | |
| Yes (%) | 383 (33.9) | 44 (44.0) | 339 (32.9) | |
| Placed on CPAP | | | | 0.14† |
| N | 1118 | 99 | 1019 | |
| Yes (%) | 23 (2.1) | 4 (4.0) | 19 (1.9) | |
| Outcomes | | | | |
| Length of hospital stay, days | | | | 0.21* |
| N | 1086 | 96 | 990 | |
| Median (IQR) | 2.0 (2.0–4.0) | 2.0 (2.0–4.0) | 2.0 (2.0–3.0) | |
| Mortality | | | | 0.07† |
| N | 1139 | 101 | 1038 | |
| Yes (%) | 5 (0.4) | 2 (2.0) | 3 (0.3) | |

*Wilcoxon rank-sum test.
†Fisher's exact test.
‡Independent two sample t-test.
CPAP, continuous positive airway pressure; HEU, HIV-exposed, uninfected; HUU, HIV-unexposed, uninfected; IMCI, Integrated Management of Childhood Illness; IQR, interquartile range; MUAC, mid-upper arm circumference; RISC, Respiratory Index of Severity in Children.

This study has several limitations. Only a very small proportion of HIV-exposed infants had HIV status tested by PCR, as recommended by national guidelines. This was due to shortage of HIV counsellors, lack of testing reagents and reluctance of parents to obtain additional testing on their children. We did not recruit moribund children who died very soon after arrival, and therefore, we may have underestimated our fatality rate. Demographics may play into admissions at a government referral tertiary hospital, and therefore, there may be selection bias for sicker children to present. A substantial proportion of recruited participants had incomplete files and were therefore not included in the analysis. Due to small numbers, we were unable to calculate predictive values for severe outcomes. However, this was a detailed prospective study conducted in a low resource setting in the post-PCV era where robust diagnostic testing was available.

In conclusion, in this setting where prevalence of HIV and malnutrition is high, children hospitalised fulfilling the WHO IMCI criteria for clinical pneumonia in the post-PCV era present with heterogeneous features. These vary by HIV exposure status but this does not influence either the frequency of danger signs or mortality. The sample size limited our ability to calculate predictive scores; and the performance of available severity scores in this population and the absence of more specific diagnostics hinder appropriate antimicrobial stewardship and the rational application of other interventions. Further work in developing antimicrobial stewardship scores in

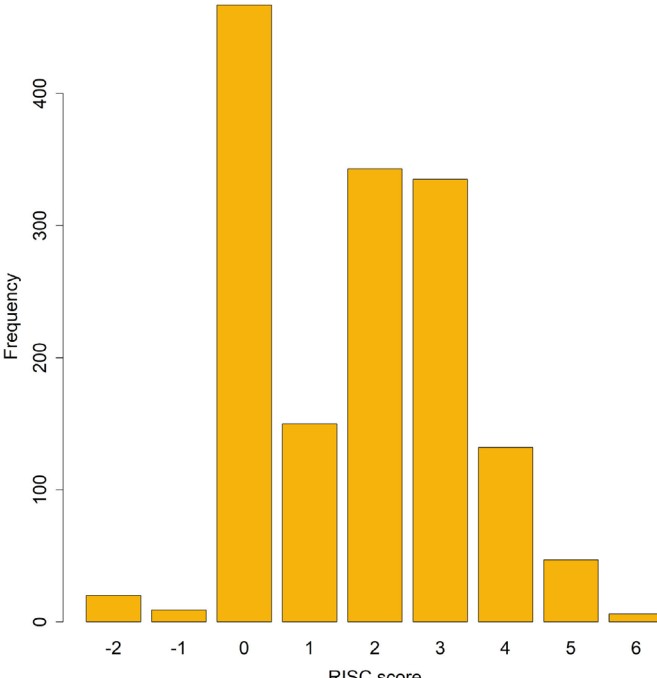

**Figure 1** Summary of RISC scores computed from the data. RISC, Respiratory Index of Severity in Children.

resource-limited settings, in a period defined by rising rates of antimicrobial resistance, may be warranted.

**Author affiliations**
[1]Department of Paediatrics and Child Health, Queen Elizabeth Central Hospital, Blantyre, Malawi
[2]Department of Clinical Sciences, Liverpool School of Tropical Medicine, Liverpool, UK
[3]Paediatrics and Child Health Research Group, Malawi-Liverpool Wellcome Programme, Blantyre, Malawi
[4]Statistical Support Unit, Malawi-Liverpool Wellcome Programme, Blantyre, Malawi
[5]Clinical Infection, Microbiology and Immunology, University of Liverpool, Liverpool, UK
[6]Pathology and Infectious Diseases, Khalifa University, Abu Dhabi, UAE
[7]Government of Malawi Ministry of Health, Lilongwe, Malawi
[8]Centre for Global Vaccine Research, Institute of Infection and Global Health, University of Liverpool Faculty of Health and Life Sciences, Liverpool, UK
[9]Division of Infection and Immunity, University College London, London, UK
[10]Global Disease Epidemiology and Control, Johns Hopkins University School of Public Health, Baltimore, Maryland, USA

**Acknowledgements** The authors wish to thank the collaborating members of the VacSurv Consortium (James Beard, Amelia C Crampin, Carina King, Sonia Lewycka, Hazzie Mvula, Tambosi Phiri, Miren Iturriza-Gomara, Osamu Nakagomi, Jennifer Verani, Cynthia Whitney).

**Collaborators** VacSurv Consortium: James Beard, Amelia C Crampin, Carina King, Sonia Lewycka, Hazzie Mvula, Tambosi Phiri, Miren Iturriza-Gomara, Osamu Nakagomi, Jennifer Verani, Cynthia Whitney.

**Contributors** Conceived and designed the research question: PI and NBZ; Conducted the statistical analysis: JC and MH; Managed the study: LN and IM; Developed and awarded funding for the study: DE, NC, CM, NF and RSH; Wrote the first draft of the manuscript: PI; Provided critical feedback on the manuscript: JC, MH, NC, NF, RSH and NBZ; Reviewed and approved the final draft: all authors. PI and NBZ accept full responsibility for the work and/or the conduct of the study, had access to the data, and controlled the decision to publish.

**Funding** This work was supported by the CDC through a cooperative agreement (grant 5U01CK000146-04), Wellcome Trust Programme Grant (Grant number 091909/Z/10/Z) and the MLW Programme Core Award (Grant 206454) from the Wellcome Trust.

**Disclaimer** The funders had no role in study design, data collection and analysis, decision to publish, or preparation of the manuscript. The findings and conclusions in this report are those of the authors and do not necessarily represent the official position of the Centers for Disease Control and Prevention, Atlanta USA.

**Competing interests** PI has received investigator-initiated research grant support from Bill & Melinda Gates Foundation outside the scope of this work; NC has received research grant support and honoraria for participation in rotavirus vaccine advisory board meetings from GlaxoSmithKline Biologicals and from Takeda Pharmaceuticals outside the scope of this work; NF has received investigator-initiated research grant support from GlaxoSmithKline Biologicals and from Takeda Pharmaceuticals outside the scope of this work. NBZ has received investigator-initiated research grant support from GlaxoSmithKline Biologicals, Takeda Pharmaceuticals, Merck-Sharpe-Dohme and the Serum Institute of India, all outside the scope of this work. All other authors declare that they have no financial disclosures or competing interests.

**Patient consent for publication** Not applicable.

**Ethics approval** This study was approved by the University of Malawi College of Medicine Research Ethics Committee (P.05/13/1386) and by the University of Liverpool (RETH490). Participants gave informed consent to participate in the study before taking part.

**Provenance and peer review** Not commissioned; externally peer reviewed.

**Data availability statement** Data are available on reasonable request. Data is available on reasonable request to the corresponding author.

**ORCID iD**
Pui-Ying Iroh Tam http://orcid.org/0000-0002-3682-8892

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
