## [Reviewer comments · BMJ Open]

ARTICLE DETAILS

TITLE (PROVISIONAL)	Clinical pneumonia in the hospitalised child in Malawi in the post-pneumococcal conjugate vaccine era: a prospective hospital-based observational study
AUTHORS	Iroh Tam, Pui-Ying; Chirombo, James; Henrion, Marc; Newberry, Laura; Mambule, Ivan; Everett, Dean; Mwansambo, Charles; Cunliffe, Nigel; French, Neil; Heyderman, Robert; Bar-Zeev, Naor

VERSION 1 – REVIEW

REVIEWER	Hashmi, S. Shahrukh University of Texas Health Science Center Houston
REVIEW RETURNED	18-May-2021

GENERAL COMMENTS	This was a well-written and presented manuscript. Although the authors did achieve their aims - describing clinical pneumonia in hospitalized children in the post-PCV era - I was left feeling that there may have been more that they could have described. The RISC scores performed poorly in the study population as reported by the authors. However, what now? Was there anything else in their data that was more informative? I think the manuscript would benefit with some discussion along those lines (if the data permits). Some specific questions/comments are: 1. Please add IQR for median age (line 191)2. In the paragraph starting on line 194, the authors mention the clinical features that were present. Are these features those that were present at baseline? Or were they features that the patient presented with at any time during their follow-up in the study? Please specify.3. Line 203-4: Please add IQR for median ages for the HEU vs HUU comparison.4. Are the ages presented in the manuscript, raw chronological age based on time since birth? Or are they adjusted ages to account for prematurity if present? Since data was collected in a prospective manner and if so, how were repeat measures (if any) incorporated? Please add more information in Methods to assist in reproducibility of the study design.5. What was the prevalence of malaria in the cohort? How many patients were comorbid with both HIV and malaria? Did they have the highest RISC scores?
---

	6. Do the authors feel that the large difference in sample size and in some cases variability between the HEU and HUU patients, influences the results of the tests they used? 7. The authors do present how the HEU and HUU patients differed from each other with respect to clinical presentation and RISC scores. However, there is limited info presented on how the RISC scores in these two group related to clinical outcomes? The authors said that the scores were not very predictive, but the data for that is not presented. I realize that the small numbers might have precluded predictive values for severe outcomes, but some information related to those (despite the imprecision in the estimates) would be beneficial to the reader. 8. The cohort was made up of hospitalized pneumonia patients. Among them, the highest RISC scores were more common in HEU patients. However, the outcomes were comparable between the HEU and HUU patients (possibly due to improvements in treatment of HIV patients, as mentioned by the authors). What role, if any, do the authors think selection bias played in these results? Are there demographic or social reasons whereby local hospital admission rates may differ in the two groups (independent on RISC scores)? How does the prevalence of HIV in the study cohort (10%) compare with HIV in the pediatric population? 9. How much of a role did RISC scores play in hospitalization of these patients? I'm assuming that it was possibly not directly influencing the admissions, rather the clinicians were basing their decision to admit based on clinical features (that may also be assessed by RISC). Again, I'm trying to get at potential selection biases that might be playing a role. 10. The authors had written that "In our cohort, the lack of predominating features of clinical pneumonia points to another possibility that, in post-PCV settings where HIV and chronic malnutrition rates are high (10% and 39%, respectively), children do not present with typical signs and symptoms." Did the authors interrogate the data to assess if there are any atypical signs/symptoms that might be better at predicting in the post-PCV setting in the population under study? Potentially something that could replace the current scoring tools? 11. Due to various differences between the HEU and HUU groups, and the stark difference in sample sizes, did the authors consider further analyses to adjust for the differences? Is it the HIV status or the various other clinical or demographic differences between these patients that are influencing the differences in RISC? Or perhaps a combination thereof. The authors do make a clear presentation on how the present scoring tool does not have sufficient discriminating ability with respect to the new post-PCV setting. However, I was wondering if there are any insights they got from this large dataset that can suggest an improved system to predict clinical outcomes?
--	---

REVIEWER	Chan, Jocelyn Murdoch Childrens Research Institute Centre of Research Excellence in Newborn Medicine
REVIEW RETURNED	15-Jun-2021

GENERAL COMMENTS	This is an important and well-conducted observational study describing the clinical characteristics of children with pneumonia in Malawi post-PCV introduction. The usefulness of the study could be strengthened by comparing concordance of the two available criteria for categorising pneumonia severity (RISC and IMCI) and inclusion of data on antibiotic usage within this cohort. Abstract  • Abstract should add some minimal information about the study setting • MUAC should be spelt out and it would be helpful to readers to provide some indication of normal to assist with interpretation of malnutrition scores. • IMCI should also be spelt out. • Suggest including clinical outcomes (i.e. deaths) in addition to reporting similarity of clinical outcomes between groups. Introduction  • Would be useful clarify for readers how RISC scores (or other scores) can be used to improve antimicrobial stewardship. Are they currently used in Malawi or any other context in this way? • Would also be useful to include a brief description of the components of the RISC score. • Are IMCI danger signs used as part of clinical practice and in what way? Methods  • First para should also include age eligibility • Suggest including the WHO IMCI criteria • Line 185 – broken link Results  • Were there any systematic differences between children included in the analysis and the 521 with incomplete files? • Any missing data for the 1139 children in the analysis? What were the rates of follow-up? Consider including discussion of limitations if required. • As above, suggest reporting vaccination status since this can affect disease severity. • Was there data available on the antibiotic therapy administered? E.g. IV vs. oral antibiotics? Broad spectrum or narrow spectrum? This would be relevant to report given your background discusses antibiotic stewardship. • Recommend reporting the concordance between WHO IMCI categorisation of pneumonia severity against RISC scores • Did the deaths occur during admission or post admission (ascertained during follow-up)?  o What were the RISC scores for the five children who died? Did they belong to the highest RISC score category (3+)? Discussion  • Line 229 – You mention differences in age and anthropometry. Are you referring to differences between HEU and HUU groups? Is this data available to clinicians and does it affect clinical management of pneumonia? Please clarify. • Line 241 – Could you please expand on the uneven distribution and divergent results of the risk scores? Divergent to what? Uneven in what way? What distribution are you expecting and why? • Line 277 – I'm not sure you have provided adequate evidence of poor performance of the risk score in this population? Please clarify?
--

VERSION 1 – AUTHOR RESPONSE

Reviewer: 1

Dr. S. Shahrukh Hashmi, University of Texas Health Science Center Houston

Comments to the Author:

This was a well-written and presented manuscript. Although the authors did achieve their aims - describing clinical pneumonia in hospitalized children in the post-PCV era - I was left feeling that there may have been more that they could have described. The RISC scores performed poorly in the study population as reported by the authors. However, what now? Was there anything else in their data that was more informative? I think the manuscript would benefit with some discussion along those lines (if the data permits).

Author response: We thank the reviewer for the comments and agree that the RISC scores performed poorly. We did review the data extensively and conducted several thorough analyses to see what other points could be extracted from our data that aligned with the main research questions. However, further analyses of the data beyond our research questions fell outside of the scope of this paper. In the discussion, we do review the variety of existing pneumonia risk scores, consider why the risk scores did not perform well, and consider the implications of this for antimicrobial stewardship.

Some specific questions/comments are:

1. Please add IQR for median age (line 191)

Author response: This has been added [L57, L218].

2. In the paragraph starting on line 194, the authors mention the clinical features that were present. Are these features those that were present at baseline? Or were they features that the patient presented with at any time during their follow-up in the study? Please specify.

Author response: We have clarified this to clinical features present at enrollment [L222].

3. Line 203-4: Please add IQR for median ages for the HEU vs HUU comparison.

Author response: This has been added [L233].

4. Are the ages presented in the manuscript, raw chronological age based on time since birth? Or are they adjusted ages to account for prematurity if present? Since data was collected in a prospective manner and if so, how were repeat measures (if any) incorporated? Please add more information in Methods to assist in reproducibility of the study design.

Author response: Age is presented based on verbal and written documentation from parents on time since birth. These details have been added to the methods [L173-174].

5. What was the prevalence of malaria in the cohort? How many patients were comorbid with both HIV and malaria? Did they have the highest RISC scores?

Author response: In the region, the prevalence of malaria is documented to be 36% [Chilanga Malaria Journal 2020]. However, although malaria testing is done as part of routine clinical care, test results were not collected in this cohort, and therefore HIV-malaria comorbidity data is not available.

6. Do the authors feel that the large difference in sample size and in some cases variability between the HEU and HUU patients, influences the results of the tests they used?

Author response: We believe that although our sample size of over 1,100 children included in the analysis is large, we would have of course benefited from a larger sample size, as well as larger numbers of HEU children. We did mention sample size as a limitation in our discussion [L309-310].

7. The authors do present how the HEU and HUU patients differed from each other with respect to clinical presentation and RISC scores. However, there is limited info presented on how the RISC scores in these two group related to clinical outcomes? The authors said that the scores were not very predictive, but the data for that is not presented. I realize that the small numbers might have precluded predictive values for severe outcomes, but some information related to those (despite the imprecision in the estimates) would be beneficial to the reader.

Author response: We thank the reviewer for this comment and in fact had initially planned to include more analyses than was eventually submitted. However, we decided not to, for several reasons: 1) the relatively small numbers of HEU children as mentioned; 2) the limited generalizability and interpretation of the statistical analysis findings; and 3) the clinical relevance and applicability of the findings to the general lay reader. We wanted the paper to be able to be understood by the general clinician or health policy maker, and felt that including these information would not be robust, relevant, or meaningful.

8. The cohort was made up of hospitalized pneumonia patients. Among them, the highest RISC scores were more common in HEU patients. However, the outcomes were comparable between the HEU and HUU patients (possibly due to improvements in treatment of HIV patients, as mentioned by the authors). What role, if any, do the authors think selection bias played in these results? Are there demographic or social reasons whereby local hospital admission rates may differ in the two groups (independent on RISC scores)? How does the prevalence of HIV in the study cohort (10%) compare with HIV in the pediatric population?

Author response: Selection bias is always a concern. The hospital where the study took place is a government tertiary referral hospital for the southern region of the country, and a site providing HIV care, and therefore one may expect higher than usual rates of hospital admissions among HEU and/or HIV-infected children. In our case, the prevalence of HIV in the study cohort was equivalent to national levels (HIV prevalence ~10%).

9. How much of a role did RISC scores play in hospitalization of these patients? I'm assuming that it was possibly not directly influencing the admissions, rather the clinicians were basing their decision to admit based on clinical features (that may also be assessed by RISC). Again, I'm trying to get at potential selection biases that might be playing a role.

Author response: Routine clinical care and decision for hospital admission is primarily based on clinician judgement. Although RISC scores are available, and clinicians will consider RISC components in their decision-making, such as HIV status, O2 levels, etc, RISC scores are not calculated or formally used in this clinical setting. Instead, RISC scores were calculated after hospitalization for the purposes of this study.

10. The authors had written that "In our cohort, the lack of predominating features of clinical pneumonia points to another possibility that, in post-PCV settings where HIV and chronic malnutrition rates are high (10% and 39%, respectively), children do not present with typical signs and symptoms." Did the authors interrogate the data to assess if there are any atypical signs/symptoms that might be

better at predicting in the post-PCV setting in the population under study? Potentially something that could replace the current scoring tools?

Author response: We considered that, but were limited by the data that were collected during the study. These variables were decided prior to study initiation and listed only signs and symptoms that one might reasonably expect to encounter in someone with a diagnosis of pneumonia.

11. Due to various differences between the HEU and HUU groups, and the stark difference in sample sizes, did the authors consider further analyses to adjust for the differences? Is it the HIV status or the various other clinical or demographic differences between these patients that are influencing the differences in RISC? Or perhaps a combination thereof. The authors do make a clear presentation on how the present scoring tool does not have sufficient discriminating ability with respect to the new post-PCV setting. However, I was wondering if there are any insights they got from this large dataset that can suggest an improved system to predict clinical outcomes?

Author response: We appreciate the reviewer efforts for us to get as much out of this dataset as possible! In fact, we were extremely keen to identify a better predictor of pneumonia severity – but the reality is that the sample size, and the dataset, was what it was. We considered multiple additional analyses, and in fact pursued other statistical analyses as mentioned above. However, ultimately, we felt for the findings to be robust, relevant, and meaningful, we should adhere to and report findings on the main research questions that we posed.

Reviewer: 2

Dr. Jocelyn Chan, Murdoch Childrens Research Institute Centre of Research Excellence in Newborn Medicine

Comments to the Author:

This is an important and well-conducted observational study describing the clinical characteristics of children with pneumonia in Malawi post-PCV introduction. The usefulness of the study could be strengthened by comparing concordance of the two available criteria for categorising pneumonia severity (RISC and IMCI) and inclusion of data on antibiotic usage within this cohort.

Abstract

- Abstract should add some minimal information about the study setting

Author response: We included additional information on the study setting [L51].

- MUAC should be spelt out and it would be helpful to readers to provide some indication of normal to assist with interpretation of malnutrition scores.

Author response: This was done; however, there was insufficient space in the abstract to include normal ranges (which would be normal <11.5 cm).

- IMCI should also be spelt out.

Author response: This was done [L70].

- Suggest including clinical outcomes (i.e. deaths) in addition to reporting similarity of clinical outcomes between groups.

Author response: We included mortality outcomes [L65].

Introduction

- Would be useful clarify for readers how RISC scores (or other scores) can be used to improve antimicrobial stewardship. Are they currently used in Malawi or any other context in this way?

Author response: Pneumonia severity scores do not currently inform antimicrobial stewardship; however, they reflect more generally the nature of antimicrobial prescribing, which in low resource settings are based more on the perception of pneumonia severity rather than an abundance of objective data. We clarified how risk scores can impact antimicrobial use and stewardship [L146-147].

- Would also be useful to include a brief description of the components of the RISC score.

Author response: The components of the RISC score are listed in Table 1 [L448].

- Are IMCI danger signs used as part of clinical practice and in what way?

Author response: Clinicians are taught the IMCI danger signs as part of their training, and rely on these danger signs to decide how to manage the patient and whether to escalate care.

Methods

- First para should also include age eligibility

Author response: The age eligibility is listed in the first paragraph [L173]. As this study was nested within a larger PCV13 effectiveness study, all children old enough and eligible to receive the PCV13 were enrolled into the parent study, and enrolled into this study if they were admitted with a diagnosis of pneumonia.

- Suggest including the WHO IMCI criteria

Author response: We included a reference to WHO IMCI treatment guidelines [L194].

- Line 185 – broken link

Author response: Thank you for raising this to our attention. This link actually refers to Table 1, and must be a formatting error when we put together the submission on Editorial Manager. We have corrected the link.

Results

- Were there any systematic differences between children included in the analysis and the 521 with incomplete files?

Author response: Key clinical features were computed for the excluded children and showed that there were no systematic differences between children included in the analysis and those excluded.

- Any missing data for the 1139 children in the analysis? What were the rates of follow-up? Consider including discussion of limitations if required.

Author response: The 1,139 children included in the analysis had complete files. In the HEU group, rate of follow-up was 90% while in the HUU group, the rate of follow-up was 89%.

- As above, suggest reporting vaccination status since this can affect disease severity.

Author response: As this study was nested within a prospective study of PCV13 effectiveness [L174], all children hospitalized with pneumonia had received at least one dose of PCV13.

- Was there data available on the antibiotic therapy administered? E.g. IV vs. oral antibiotics? Broad spectrum or narrow spectrum? This would be relevant to report given your background discusses antibiotic stewardship.

Author response: We agree with the reviewer that these data are relevant. Clinicians provided antibiotic therapy per department guidelines (first-line being penicillin and gentamicin, second-line being ceftriaxone), but data on the specific antibiotic therapy administered for each patient were not captured.

- Recommend reporting the concordance between WHO IMCI categorisation of pneumonia severity against RISC scores

Author response: We did not have a sufficiently large sample size or mortality to calculate predictive values and compare concordance between IMCI danger signs and RISC. However, Table 2 does show whether differences between HEU and HUU are statistically significant, for HEU/HUU children with at least one IMCI danger sign, and by RISC score [L471].

- Did the deaths occur during admission or post admission (ascertained during follow-up)?

Author response: Two deaths were recorded as occurring during the admission period (1 HEU and 1 HUU). The other dates of the other deaths were not recorded.

- o What were the RISC scores for the five children who died? Did they belong to the highest RISC score category (3+)?

Author response: Four of the children who died had RISC scores in the lowest category. Only one child had a RISC score in the highest category.

Discussion

- Line 229 – You mention differences in age and anthropometry. Are you referring to differences between HEU and HUU groups? Is this data available to clinicians and does it affect clinical management of pneumonia? Please clarify.

Author response: We are referring HEU children being significantly younger and having significantly lower MUAC than HUU children, and thereby being potential indicators to triage in clinical management of pneumonia. We have clarified the language [L258-260].

- Line 241 – Could you please expand on the uneven distribution and divergent results of the risk scores? Divergent to what? Uneven in what way? What distribution are you expecting and why?

Author response: We expected a parametric distribution based on assumptions that data would be normally distributed, but Fig1 was not and clearly diverged from a normal distribution.

- Line 277 – I'm not sure you have provided adequate evidence of poor performance of the risk score in this population? Please clarify?

Author response: The reviewer is correct, we did not have sufficient numbers to calculate predictive values for severe outcomes. We have clarified the language [L309-310].

VERSION 2 – REVIEW

REVIEWER	Hashmi, S. Shahrukh University of Texas Health Science Center Houston
REVIEW RETURNED	09-Sep-2021

GENERAL COMMENTS	The authors have revised the manuscript based on the comments by both reviewers. Although they responded to all comments, some explanations were not mirrored with edits or details added to the manuscript. However, the manuscript is much improved product. A few comments:  1. The authors discussed the presence of selection bias in response to the 1st reviewer's comment. However, they don't touch on how demographics may play into admissions at a government tertiary referral hospital . Please add a sentence about selection bias and demographics in the discussion section (with the other limitations). Information about the hospital and the demographics of the area (and selection bias in admissions) is informative for readers who may not be aware of the Malawi population under study. 2. Line 231. Please explain which "clinical outcomes" are being referred to or add a reference to Table 2 after the sentence "Clinical outcomes between the HEU and HUU groups were also similar." 3. The authors responded to the 2nd reviewer's comment in the sentence on RISC scores now listed on line 262 (previously line 241). However, the sentence in question has not been edited. Please edit for clarity (e.g. use "not normally distributed" rather than "unevenly distributed" if that is what was meant, and similar edits to describe "divergent"). 4. In Table 2, for RISC score, please change the category from "-2,-3" to "-2 to +2" or "-2 to +3" as applicable. Also, "3" is listed in both categories. Please make the categories mutually exclusive. 5. The word "respectively" can be deleted from the abstract - line 61.
---

VERSION 2 – AUTHOR RESPONSE

Reviewer Comments to Author:

1. The authors discussed the presence of selection bias in response to the 1st reviewer's comment. However, they don't touch on how demographics may play into admissions at a government tertiary referral hospital . Please add a sentence about selection bias and demographics in the discussion section (with the other limitations). Information about the hospital and the demographics of the area (and selection bias in admissions) is informative for readers who may not be aware of the Malawi population under study.

Author response: This has been included in the limitations [L278-279].

2. Line 231. Please explain which "clinical outcomes" are being referred to or add a reference to Table 2 after the sentence "Clinical outcomes between the HEU and HUU groups were also similar."

Author response: We added a reference to Table 2 [L221].

3. The authors responded to the 2nd reviewer's comment in the sentence on RISC scores now listed on line 262 (previously line 241). However, the sentence in question has not been edited. Please edit for clarity (e.g. use "not normally distributed" rather than "unevenly distributed" if that is what was meant, and similar edits to describe "divergent").

Author response: We edited to 'not normally distributed' [L251].

4. In Table 2, for RISC score, please change the category from "-2,-3" to "-2 to +2" or "-2 to +3" as applicable. Also, "3" is listed in both categories. Please make the categories mutually exclusive.

Author response: This has been clarified [Table 2].

5. The word "respectively" can be deleted from the abstract - line 61.

Author response: We removed the word [L59].